# Implementation of Cellulose-Based Filtration Aids in Industrial Sunflower Oil Dewaxing (Winterization): Process Monitoring, Prediction, and Optimization

**DOI:** 10.3390/foods13182960

**Published:** 2024-09-19

**Authors:** Tanja Lužaić, Katarina Nedić Grujin, Lato Pezo, Branislava Nikolovski, Zoran Maksimović, Ranko Romanić

**Affiliations:** 1Faculty of Technology Novi Sad, University of Novi Sad, Bulevar cara Lazara 1, 21000 Novi Sad, Serbia; tanja.luzaic@tf.uns.ac.rs (T.L.); katarina.nedic.grujin@gmail.com (K.N.G.); barjakb@uns.ac.rs (B.N.); 2Dijamant Ltd., Temišvarski drum 14, 23000 Zrenjanin, Serbia; 3Institute of General and Physical Chemistry, University of Belgrade, Studentski trg 12/V, 11158 Belgrade, Serbia; latopezo@yahoo.co.uk; 4Faculty of Pharmacy, University of Belgrade, Vojvode Stepe 450, 11221 Belgrade, Serbia; zmaksim1@pharmacy.bg.ac.rs

**Keywords:** sunflower oil, dewaxing, refining, cellulose-based filtration aid, optimization

## Abstract

In the production of refined sunflower oil, waxes are removed during the winterization stage, and wax crystals are separated through filtration assisted by filtration aids. Commonly used filtration aids in oil refining include perlite and diatomaceous earth. After winterization, a significant amount of filter cake remains as a by-product and is treated as waste. Today, natural cellulose fibers are being promoted as filtration aids. Their advantages are numerous, both in the production process and from an environmental perspective. However, their only disadvantage is their higher cost. Therefore, in this study, 57 filtration cycles during the industrial sunflower oil winterization step using cellulose-based filtration aids were monitored. Different process parameters, including the pressure differential on the filter, the flow rate of filtered oil, constant pressure period, the quantity of filtered oil, filtration time, the quantity of pre-coating and dosing filtration aids, the volume of filtered oil, the concentration of dosing filtration aid, as well as the mass of separated waxes, were observed. Additionally, artificial neural networks were applied to predict process parameters, optimize the process, and, above all, determine the dosage of filtration aids, which will make the process more economical. The optimal filtration process is performed at a pressure differential of 3.3 bar, lasting a total of 39 h, with 32 h at constant pressure, resulting in 322,503 kg of filtered oil and 90.41 kg of waxes. The optimal quantity of cellulose-based filtration aids employed for pre-coat was 80 kg, and for dosing, 375 kg, with an optimal concentration of 0.12% *w/w*.

## 1. Introduction

Edible oil refining is used to improve the quality and safety of edible oils, making them suitable for consumption. The main aims of refining are to remove impurities, enhance flavor and odor, improve stability, and ensure safety [1]. Impurities present in crude oil have different physical and chemical characteristics, and different procedures have been developed for their removal. These processes include degumming, neutralization, bleaching, winterization, and deodorization [2,3]. Winterization (dewaxing) is a crucial step in oil refining, specifically aimed at removing waxes and triacylglycerols with higher melting points. Waxes are considered undesirable components due to their propensity to crystallize at room temperature, causing cloudiness in refined oils [4,5]. In the case of sunflower oil, dewaxing is particularly important because of its high wax content compared to other vegetable oils [6]. The traditional dewaxing method involves gradual crystallization with gentle mechanical mixing and subsequent wax separation through filtration using appropriate filters [1,7]. The crystallization temperature is typically below 15 °C, with a minimum retention time of 4 h [2,3]. In addition to the traditional method, an alternative winterization process using solvents, most commonly *n*-hexane, is employed. The solvent reduces the viscosity of the oil, thereby facilitating the crystallization of waxes. The filtration process is conducted at lower temperatures (between −5 °C and −7 °C), which leads to a threefold cost reduction [7,8]. Another innovative approach is the “wet” winterization, which is suitable for crude oils with higher wax content. In this method, the oil is cooled, and a specific amount of low-concentration sodium hydroxide solution is added to increase soap content. Agglomeration of waxes around soap particles occurs in crystallizers equipped with mixers. After standing, the mixture is transferred to another crystallizer, where a small amount of chilled water is added. Following aging, the oil is heated using a heat exchanger and separated using a centrifugal separator. However, post-separation cloudiness is common. To further reduce wax content, the oil is washed with cold water after separation, resulting in an additional 10–20 mg/kg reduction in wax content.

Filtration during winterization differs from other refining stages due to the specific nature of waxes. Solid particles causing oil cloudiness are often small and compressible, leading to pore blockage and filtration cycle interruption. Filtration aids, such as perlite and diatomaceous earth, are commonly used to enhance efficiency [9]. This aid should not affect the oil’s chemical composition, aroma, or taste and should consist of solid, finely porous, non-compressible particles. Prior to filtration, a filter aid is applied to the porous filter media as a pre-coat layer. During filtration, it is continuously dosed into the oil before the filter. This significantly reduces the resistance of the filtration layer and ensures both high flow rates and efficient separation [10,11]. Filtration aids primarily consisting of natural cellulose fibers are being promoted worldwide [12]. The advantages of these filtration aids are numerous. They have a low specific mass, which reduces their consumption and reduces oil losses with the filter cake. These aids have a low fiber density that allows them to be easily suspended in oil without sedimentation [13]. After filtration, the filter cake remains. Empirical data from refined oil manufacturers indicate that approximately 4 kg of filter cake is generated per ton of refined oil. If perlite and diatomaceous earth, which are commonly used in Serbia, are used as filtration aids, the resulting cake is almost useless. It is typically treated as waste, either discarded in landfills or used for combustion. According to the Statistical Office of the Republic of Serbia data, the annual production of refined sunflower oil in Serbia averages between 160,000 and 180,000 tons, generating approximately 640 to 720 tons of filter cake, practically that amount of “waste”. Using cellulose-based filtration aids, the range of applications for filter cake could be expanded. According to recommendations from cellulose-based filtration aid manufacturers, filter cake could be used in livestock feed production because it does not contain toxic substances or silica crystals. Another advantage is that cellulose fibers do not cause wear on filters, pumps, or filtration equipment. Their fibrous mesh structure facilitates the removal of filter cake from the filter. These aids have a low fiber density that allows them to be easily suspended in oil without settling. Cellulose-based filtration aids provide a finer filter media matrix to trap fine solids and ensure filtrate clarity [14,15].

Nedić Grujin et al. (2023) [5] demonstrated that cellulose-based filtration aids, used in the winterization stage, are effective in removing waxes and also impact the reduction of phospholipids, free fatty acids, soaps, pigments, and heavy metals from sunflower oil. The only drawback of these filtration aids is their higher cost compared to conventional filtration aids. Therefore, the objective of this study is to monitor filtration process parameters during the industrial sunflower oil winterization step, predict process parameters, optimize the process, and, above all, determine the optimal dosage of filtration aids.

## 2. Materials and Methods

### 2.1. Industrial Winterization (Dewaxing)

This study focused on the filtration of sunflower oil during industrial winterization. Winterization is a refining stage aimed at removing waxes from the oil. Prior to winterization, oil neutralization was performed through acid degumming, followed by oil deodorization. The industrial refining capacity was 200 tons of crude oil per day. The process flow diagram for the winterization stage is shown in Figure 1. During oil filtration, specialized cellulose-based filtration aids were used for both pre-coating and dosing. The filter cake was separated using two horizontal pressure leaf filters (Amafilter BV, Alkmaar, The Netherlands) after drying the filters with compressed air for 15 min. The oil temperature during filtration was 16 °C. The filter volume was 6880 L, with a total of 30 filter leaves. Each leaf had a surface area of 2 × 1 m^2^, resulting in a total filtration area of 60 m^2^. Two filters were employed and operated alternately.

### 2.2. Cellulose-Based Filtration Aids

For filtration in this study, cellulose-based filtration aids were used. One was labeled FA-P and used for pre-coating, while two different filtration aids, labeled FA-I and FA-II, were used for dosing during filtration. FA-P was a white powder with particle granulation ranging from 5 to 50%, >100 µm, and at least 45%, >32 µm, with a bulk density of 120 to 200 kg/m^3^. FA-I had particle granulation up to 70%, >100 µm, and up to 98%, >32 µm, with a bulk density of 160 to 220 kg/m^3^. FA-II contained particles with granulation up to 55%, >100 µm, and up to 95%, >32 µm, with a bulk density of 170 to 240 kg/m^3^ (JRS, J. Rettenmaier & Sohne GMBH, Rosenberg, Germany). Before filtration, an initial filtration layer (pre-coat) was applied, formed by evenly depositing a suspension of a specific amount of pre-coating filtration aid (FA-P) and oil onto the filter leaves using a pump. Once the pre-coating process was complete, filtration of the oil proceeded in parallel with dosing a suspension of filtration aids (FA-I or FA-II) into the oil. Filtration was interrupted when the filter’s working pressure reached a maximum of 3.6 bar or when the total amount of filtration aids (both pre-coating and dosing aids) used for filtration exceeded 460 kg. The study varied the amount of pre-coating aid (dosages of 60, 80, 100, and 125 kg) and dosing aid (added between 150 and 450 kg of FA-I or FA-II), with only one dosing aid used in each filtration cycle. The selected aid quantities aimed to achieve concentrations of 0.55%, 0.73%, and 0.91% based on empirical findings from previous use of perlite as a filtration aid. Larger variations in added filtration aid quantities were not considered due to the industrial-scale experiment’s economic constraints.

### 2.3. Oil Samples

Industrial winterization (dewaxing) was carried out from 2019 to 2020 at Dijamant Ltd. in Zrenjanin, Serbia. Oils were sampled before and after each monitored filtration cycle (a total of 57 cycles), both before and after the filter during the winterization stage. The oil samples were collected at the beginning of each filtration cycle, ranging from 0.8 to 1 L, using 1 L PET bottles sealed with original two-part closures. The obtained oil is produced from sunflower seeds (*Helianthus annuus* L.) grown in the territory of Vojvodina (north of the Republic of Serbia) in 2019.

#### Investigation of Wax Content

The wax content in sunflower oil before filtration was determined using a gravimetric method described in detail by Oštrić-Matijašević and Turkulov (1973) [16] and Nedić Grujin et al. (2023) [5]. The method is based on the selective extraction of waxes, using their characteristics to crystallize and settle in oil at lower temperatures. Separated wax crystals are first purified with *n*-hexane at 1–3 °C for 8 h, then extracted using warm alcohol for 4–6 h.

### 2.4. Monitoring of Process Parameters in Oil Filtration

During the sunflower oil filtration process within the winterization stage, the following parameters were monitored and recorded from the device screen during each filtration cycle: pressure differential on the filter, ∆p (bar), and flow rate of filtered oil, dV/dt (m^3^/h), measured every hour; constant pressure period, Τ_p const._ (h), quantity of filtered oil, FO (kg), and filtration time, FT (h), i.e., the duration of the filtration cycle, measured per filtration cycle. The amount of added filtration aid was also measured: quantity of pre-coating filtration aid, Qp (kg), and quantity of dosing filtration aids, Qd (kg).

Based on the measured data and results of oil analysis, the following parameters were calculated: volume of filtered oil, V (m^3^), calculated as the ratio of the total quantity of filtered oil FO (kg) to the oil density ρ = 920 kg/m^3^; concentration of dosing filtration aid, CS (% *w/w*), calculated as the average concentration of the dosing filtration aid in the filtered oil; and mass of separated waxes, mW (kg), determined as the mass fraction of waxes in the total quantity of filtered oil.

### 2.5. Standard Scores

The ranking of 57 samples was conducted by comparing their raw data against extreme values, utilizing the method established by Brlek et al. (2013) [17]. The criteria for ranking included the following parameters: W_content_ (mg/kg), ∆p (bar), FT (h), T_p const._ (h), ∆p (bar), dV/dt (m^3^/h), FO (kg), V (m^3^), and mW (kg), where higher values were preferred. Conversely, lower values were favored for the following parameters: Qp (kg), Qd (kg), and CS (% *w/w*).

### 2.6. Artificial Neural Network (ANN) Model

For the ANN modeling phase, which involved 57 samples, the data were split into training (60%), cross-validation (20%), and testing (20%) sets. To improve accuracy, min-max normalization was applied to standardize both input and output data. The proposed multilayer perceptron (MLP) ANN models featured three-layer architecture with a feed-forward design and utilized backpropagation training, as described in previous studies by Kavuncuoglu et al. (2017) [18], Pavlić et al. (2020) [19], and Bajić et al. (2020) [20]. The hidden layer consisted of 5 to 10 neurons, and various activation functions, including tangent, sigmoidal, exponential, and identity, were tested. The BFGS algorithm was employed to construct the ANN model, iteratively adjusting weights and biases across 100,000 different configurations to minimize the squared error, with the goal of having both the learning and cross-validation curves approach zero.

The ANN models’ accuracy was evaluated using several standard computational metrics, including the coefficient of determination (r^2^), reduced chi-square (χ^2^), mean bias error (MBE), root mean square error (RMSE), mean percentage error (MPE), sum of squared errors (SSE), and average absolute relative deviation (AARD). ANN models were used to assess the influence of dosing filtration aid, W_content_, Qp, Qd, CS, and ∆p on FT, T_p const_, dV/dt, FO, V, and mW.

#### Error Analysis

In terms of error analysis, the accuracy of the developed models was evaluated through several key metrics, i.e., coefficient of determination (r^2^), reduced chi-square (χ^2^), mean bias error (MBE), root mean square error (RMSE), mean percentage error (MPE), the sum of squared errors (SSE), and average absolute relative deviation (AARD). These widely used parameters were employed to assess the validity of the models [21]:(1)$$\chi^{2}=\frac{\sum_{i=1}^{N}(x_{\exp,i}-x_{pre,i}{)}^{2}}{N-n}$$
(2)$$RMSE={\left[\frac{1}{N}\cdot \sum_{i=1}^{N}(x_{pre,i}-x_{\exp,i}{)}^{2}\right]}^{\frac{1}{2}}$$
(3)$$MBE=\frac{1}{N}\cdot \sum_{i=1}^{N}\left(x_{pre,i}-x_{\exp,i}\right)$$
(4)$$MPE=\frac{100}{N}\cdot \sum_{i=1}^{N}\left(\frac{\left|x_{pre,i}-x_{\exp,i}\right|}{x_{\exp,i}}\right)$$
(5)$$SSE=\sum_{i=1}^{N}(x_{pre,i}-x_{\exp,i}{)}^{2}$$
(6)$$AARD=\frac{1}{N}\cdot \sum_{i=1}^{N}\left|\frac{x_{exr,i}-x_{pre,i}}{x_{exr,i}}\right|$$
where x_exp,i_ stands for the experimental values, x_pre,i_ are the predicted values calculated by the model, and N and n are the number of observations and constants, respectively.

### 2.7. Statistical Analysis

Wax content measurement was performed in triplicate (*n* = 3), while process parameters were recorded during industrial filtration, and each trial was an individual cycle. Data from 57 industrial filtration cycles were analyzed using several statistical tests, including the Kruskal–Wallis test, the Kolmogorov–Smirnov two-sample test, the Wald–Wolfowitz runs test, and the Mann–Whitney U test, to identify significant differences between samples. Principal Component Analysis (PCA) was also conducted to structure and interpret the results. All statistical analyses were performed using Statistica 10.0 (StatSoft Inc., Tulsa, OK, USA).

## 3. Results and Discussion

In this study, a total of 57 filtration cycles were monitored under industrial conditions. To initiate the filtration process, a uniformly thin layer of filter aid needs to be applied, which is distributed across the surface of each filter leaf at a rate of 0.5 to 1 kg/m^2^ [2]. The layer primarily forms to protect the filter leaves from clogging, enable the retention of fine filtrate particles, and achieve the desired clarity of the filtrate [10,11,13]. Table 1 shows the quantities of filter aids used in each filtration cycle, along with the type of filter aid used for dosing. In the industrial filtration trials, filter aid FA-P was used for pre-coat at a rate of 1.00 to 2.08 kg per m^2^ of filtration area (Table 2). A suspension was prepared for dosing at concentrations of 0.55%, 0.73%, and 0.91%. Ergönül and Nergiz (2015) [22] used diatomaceous earth and perlite as filter aids for sunflower oil winterization at concentrations of 0.30% and 0.60%, respectively. The addition of dosing filter aids (FA-I and FA-II) increases the porosity of the filter cake. The amount of added filter aid is predetermined, and the overall quantity added should be at least equal to the amount of solid phase removed during filtration [13,22].

In the conducted research, filter aid FA-I was used in 43 filtration cycles. Its specific mass varied from 2.5 to 8.3 kg/m^2^, as shown in Table 2. Filter aid FA-II was used in 14 filtration cycles, with a specific mass ranging from 4.2 to 6.7 kg/m^2^ (Table 2). The average dosing concentration for FA-I was between 0.12% *w/w* and 0.31% *w/w*, while for FA-II, it ranged from 0.10% *w/w* to 0.31% *w/w* (Table 1). The maximum concentration of filter aid FA-II at the beginning of filtration was between 0.28 and 0.36% *w/w*, while initial concentrations of FA-I ranged from 0.28% *w/w* to a final 0.56% *w/w*, consistent with previous studies [22,23].

Statistical analysis (Kolmogorov–Smirnov two-sample test, Wald–Wolfowitz runs test, Mann–Whitney U test) of the results from 57 industrial trials revealed no significant difference in the monitored parameters during filtration cycles using either filter aid FA-I or FA-II for dosing. The total quantity of dosing filter aid was not continuously added during the process; instead, the concentration gradually decreased. This trend is clearly visible in Figure 2a,c,e, which depict the change in dosing filter aid concentration during filtration cycles. In certain filtration cycles, especially those shown in Figure 2a, the decrease in dosing filter aid concentration is not evident due to filtration being stopped after around 16 h when the pressure differential reached 3.6 bar. These are samples with a high wax content, which affected the faster clogging of the pores of the filtration medium and the stopping of filtration compared to the other monitored filtration cycles where the wax content was lower.

During the observed filtrations, the pressure and flow rate on the filter changed. Initially, the pressure remained constant at the beginning of the cycle for 2 to 32 h, depending on the experiment. At the beginning of each filtration cycle, the pressure was constant (constant pressure operation) so that, afterward, it was possible to observe a period with an approximately constant filtration rate (constant rate operation). The average oil flow rate (dV/dt) per filtration cycle varied from 8.32 to 9.18 m^3^/h (Table 1), which, when expressed per unit filter area, corresponds to 0.139 to 0.153 m^3^/h/m^2^. According to Gupta (2017) [2], the recommended filtration rate should be 0.125 m^3^/h/m^2^. The amount of filtered oil obtained in one filtration cycle ranged from 57,426 to 322,503 kg, volumetrically equivalent to 62.420 to 350.547 m^3^. The amount of wax removed from the oil per filtration cycle ranged from 29.86 to 97.42 kg (Table 1) or was calculated based on the filtration area (mW/A) between 0.50 and 1.62 kg/m^2^ (Table 2). At the beginning of the cycle, pressure remained constant at 0.2 bar (0.2 × 10^5^ Pa). The maximum filtration pressure at the end of the process ranged from 3.0 to 3.6 bar (3.0 to 3.6 × 10^5^ Pa), and filtration in this pressure range was mostly interrupted when all conditions for filter stoppage were met. According to Gupta’s 2017 [2] recommendations, filtration should be stopped when the pressure drops in the range of 30–35 psi (2.07 to 2.41 × 10^5^ Pa). However, these recommendations may vary for filters of different constructions and capacities. The content of wax in the oil (W_content_) (R^2^ = 0.04; *p* = 0.74), the dosage of filtration aid (Qd) (R^2^ = −0.11; *p* = 0.42), and the average concentration of filtration aid in winterized oil (CS) (R^2^ = 0.00; *p* = 0.99) did not significantly impact the maximum filtration pressure. During periods of constant filtrate flow rate, the pressure increase per unit time varied across experiments, ranging from 1.7 to 18.1 Pa/s (Figure 2).

It has been observed that oil filtrations with lower wax content last longer, as indicated by Figure 3 and the tested correlation (R^2^ = −0.77; *p* = 0.00). Filtration cycles lasted from 7 h, observed during filtration cycle 41, to 39 h during filtration cycle 46 (Table 2). Concentrations of the filtration aid for dosing are generally slightly higher for oil filtrations with higher wax content, as previously determined by Dahlstrom et al. (1997) [13] and Ergönül and Nergiz (2015) [22]. Based on recorded parameters of the filtration process during sunflower oil winterization, differences were observed between filtration cycles of oils with different wax content. Application of the Kruskal–Wallis test revealed a significant difference among the following three sample groups: oils with low wax content (≤350 mg/kg, a total of 21 oil samples, with an average wax content of 311.57 ± 18.89 mg/kg), oils with medium wax content (≤430 mg/kg, a total of 21 oil samples, with an average wax content of 383.00 ± 24.12 mg/kg), and oils with high wax content (15 oil samples, with an average wax content of 504.93 ± 42.83 mg/kg).

During the filtration of oil with high wax content, the pressure differential began to rise after only 2 h (filtration cycles 41, 42, 44, 52), which is significantly shorter than during filtrations of oil with low wax content, where the pressure differential started to rise after 5 h of filtration at a constant pressure differential (0.2 bar), observed in filtration cycle 23 (Table 1). Additionally, in Figure 2a,b, a short filtration time at constant pressure differential is evident during cycles 12 and 34 when oils with high wax content were filtered. The maximum filtration time for oil with low wax content was 32 h in filtration cycle 46 (Figure 2e,f). On average, the filtration time at constant pressure differential for oils with low wax content was 12.24 h. For oils with medium wax content, the average filtration time at constant pressure differential was 6.43 h (with a minimum of 1 h in filtration cycle 54 and a maximum of 20 h in filtration cycle 15). Filtrations of oil with high wax content had an average filtration time of 3.87 h at constant pressure differential, with a minimum and maximum filtration time of 2 h (filtration cycles 41, 42, 44, 52) and 8 h (filtration cycle 5), respectively, as shown in Figure 2a,b. When examining oils with low wax content during filtration cycles 45 and 46, the longest filter operation times were 37 and 39 h, respectively (Figure 2f). In both filtrations, 80 kg of filtration aid was added for pre-coat, while for dosing, 275 kg of dosing aid was added in filtration cycle 45 and 375 kg in filtration cycle 46. The average concentration of dosing aid in filtration cycle 45 was 0.10%, while in filtration cycle 46, it was slightly higher at 0.12%. The filtration duration was negligibly shorter in filtration cycle 45 compared to filtration cycle 46. When 60 kg of filtration aid was added for pre-coat and 400 kg for dosing, the longest filtration time was 27 h in filtration cycle 32, with an average concentration of dosing aid of 0.19%. In filtration cycle 11, where 100 kg and 350 kg of filtration aid were added for pre-coat and dosing, respectively, the longest filtration time was 31 h (Table 1). Based on these data, it is concluded that the quantities of pre-coat and dosing filtration aids affect the filtration duration, necessitating the optimization of their individual amounts to enhance filtration efficiency.

Observing all three sample groups of oils in Figure 2, it can be inferred that a uniform reduction in the concentration of dosing filtration aid leads to more consistent changes in filtration pressure. Heertjes and Zuideveld (1978) [24], as well as Pergam et al. (2022) [25], identified the accumulation of impurities on the surface of the previously applied filtration cake during filtration. In cases of pore blockage on the filtration cake or a reduction in its porosity, the pressure differential significantly increased. To avoid this effect, the mentioned authors varied the type of filtration aid. With currently available improved theoretical and analytical models, they achieved reduced porosity solely by varying the concentration rather than uniform dosing of the filtration aid during filtration.

### 3.1. Principal Component Analysis

PCA was used to investigate the connections between different samples, as illustrated in Figure 4. The proximity of points on the PCA graph indicates similarities in patterns [26]. In the factor space, the orientation of vectors illustrates trends in variables, and their length reflects the degree of correlation strength [27]. The obtained PCA analysis indicates the grouping of samples according to wax content into low, medium, and high, as the Kruskal–Wallis test showed previously.

The first two principal components (PCs) explained a significant portion, accounting for 70.26% of the total variance in the dataset. Specifically, the first PC contributed 56.74%, while the second PC explained 13.52% of the overall variance in the collected data.

When analyzing the projection of variables onto the factor plane, notable positive contributions to the first principal component (PC1) were observed: W_content_ (10.30% based on correlation) and CS (13.68%). Conversely, FT (15.85%), T_p const._ (9.47%), FO (15.89%), V (15.89%), and mW (12.08%) had the most negative impact on the PC1 component. On the other hand, Qd (14.87%) and dV/dt (7.00%) significantly contributed positively to the second principal component (PC2), while Qp (49.12%) and Δp (22.97%) contributed negatively to the PC2 component, as shown in Figure 4.

### 3.2. Standard Scores and Process Optimization

The standard score was calculated by summing the normalized scores for each variable (W_content_, ∆p, FT, T_p const_., dV/dt, FO, V, and mW and also Qp, Qd, and CS) and then multiplying them by their respective weights (0.1, 0.067, 0.1, 0.1, 0.067, 0.067, 0.066, and 0.066 and also 0.15, 0.15, and 0.067, respectively). By maximizing the standard score function, the optimal processing parameters (W_content_, Qp, Qd, CS, and ∆p) and the optimal values for FT, T_p const_, dV/dt, FO, V, and mW were found. When the standard score function approaches a value of 1, it indicates a higher likelihood that the tested processing parameters are optimal.

According to standards score analysis (Figure 5), the optimal standard score reached the value of 0.680 for sample 46, with the following parameters: W_content_ = 281 mg/kg, Qp = 80 kg, Qd = 375 kg, CS = 0.12% *w/w*, ∆p = 3.3 bar, FT = 39 h, T_p const._ = 32 h, dV/dt = 8.99 m^3^/h, FO = 322,503 kg, V = 350.547 m^3^, and mW = 90.41 kg.

### 3.3. Artificial Neural Network

The initial assumptions about matrix parameters like biases and weights play a crucial role in shaping the artificial neural network (ANN) models’ structure and outcomes. These assumptions are vital for accurately aligning the model with the experimental data. Moreover, the models’ performance hinges on the number of neurons in its hidden layer. To counteract the influence of random correlations resulting from initial assumptions and weight initialization, the model underwent 100,000 runs with randomized topologies.

The optimal model, which achieved the highest r^2^ value, featured seven hidden neurons. Each ANN model underwent 100 epochs of training. The training accuracy showed incremental improvement with each cycle, stabilizing around the 70th to 80th epoch. Continuing training beyond 80 epochs could lead to significant overfitting, whereas 80 epochs were adequate for achieving high model accuracy without overfitting.

The optimized neural network models (Table 3) demonstrated robust generalization capabilities for the experimental data, effectively predicting outputs based on input parameters.

Using a configuration of 3–10 neurons, the ANN models achieved high r^2^ values, specifically, 0.977, 0.927, 0.401, 0.970, 0.969, and 0.940 for training. The training algorithm used was BFGS 122, with the error function being the sum of squares. The hidden activation functions were exponential, tanh, and logistic, while the output activation functions were identity, logistic, tanh, and exponential.

The models’ feature fit was assessed and presented in Table 4, showing that the ANN models exhibited a minor lack of fit tests, indicating a satisfactory prediction of the analyzed parameters. The r^2^ values for FT, T_p const._, dV/dt, FO, V, and mW predictions were 0.967, 0.924, 0.703, 0.966, 0.959, and 0.953, respectively, suggesting that the models accurately evaluated variations and fit the data well. This is also indicated by the dependence of experimental and predicted values obtained for FT, T_p const._, dV/dt, FO, V, and mW, as shown in Figure 6.

The ANN models were evaluated using multiple metrics to assess their accuracy and distribution characteristics. The ANN model for dV/dt calculation demonstrated high accuracy with a low RMSE of 0.120, indicating minimal average absolute deviation from observed values. However, it exhibited a moderately low coefficient of determination of 0.703, suggesting a moderate fit to the data. The model also displayed a left-skewed distribution (Skew = −1.796) and leptokurtic peakedness (Kurt = 8.260).

In contrast, the ANN model for FT calculation showed good accuracy with a low RMSE of 1.183 and a high r^2^ of 0.967, indicating a good fit. Its skewness (0.364) and kurtosis (0.162) values indicated a slightly right-skewed and platykurtic distribution, respectively. However, the ANN model had a moderate average absolute relative deviation of 51.474, indicating some relative deviation from observed values.

The ANN models for mW and T_p const_ calculation exhibited reasonable accuracy with moderate RMSE values of 3.744 and 1.798, respectively, and high r^2^ values (0.953 for mW and 0.924 for T_p const_). Both models showed slightly right-skewed distributions and platykurtic peakedness. However, they had higher AARD values (165.644 for mW and 81.881 for T_p const_), suggesting notable relative deviation from observed values.

The ANN models for FO and V models demonstrated lower accuracy with very high RMSE values (1.0 × 10^4^ for FO and 1.1 × 10^4^ for V). Despite high r^2^ values (0.966 for FO and 0.959 for V), they had moderately high AARD values (4.6 × 10^5^ for FO and 4.9 × 10^5^ for V), indicating significant relative deviation from observed values. These models also showed slightly right-skewed distributions and platykurtic peakedness.

In summary, the FT model exhibited the best balance of accuracy and distribution characteristics among the evaluated models, with the dV/dt model demonstrating high accuracy but with more pronounced distributional deviations.

## 4. Conclusions

The introduction of cellulose-based filtration aids in the winterization step as an integral part of sunflower oil refining, potentially addressing an environmental issue. Their use could expand the range of filter cake applications, such as livestock feed production, since they do not contain toxic substances or silica crystals. However, it is necessary to economically justify the usage of these filtration aids. In this study, the optimization of the dosing of cellulose-based filtration aids aimed to enhance process efficiency. Fifty-seven cycles of oil filtration during winterization were monitored. The filtration process effectively removed waxes from sunflower oil. A different filtration flow was observed depending on the wax content in the oil before filtration. The lowest filtration time was recorded in oil samples with high wax content and vice versa. Using artificial neural networks, process parameters were predicted. Model validation indicated good predictive capability for these parameters. Furthermore, optimization of the filtration process was conducted, resulting in the following optimal parameters: Wcontent = 281 mg/kg, Qp = 80 kg, Qd = 375 kg, CS = 0.12% *w/w*, ∆p = 3.3 bar, FT = 39 h, Tp const. = 32 h, dV/dt = 8.99 m^3^/h, FO = 322,503 kg, V = 350.547 m^3^, and mW = 90.41 kg. The advantages of using cellulose-based filtration aids in the oil winterization phase are numerous, and based on the obtained results, the process has been optimized, achieving economic justification (as shown in the Appendix A). Although the resulting filter cake can be used as animal feed, the further goal of the research is the additional valorization of the cake for the production of more economically viable products, which will be the subject of further research.

## Figures and Tables

**Figure 1 foods-13-02960-f001:**
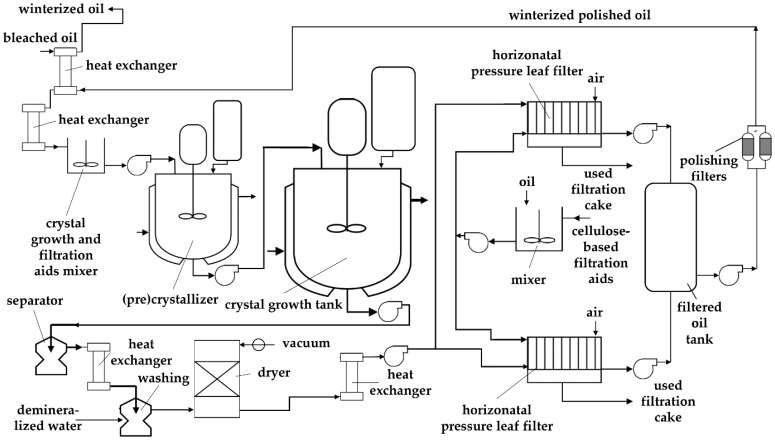
Winterization process flow diagram.

**Figure 2 foods-13-02960-f002:**
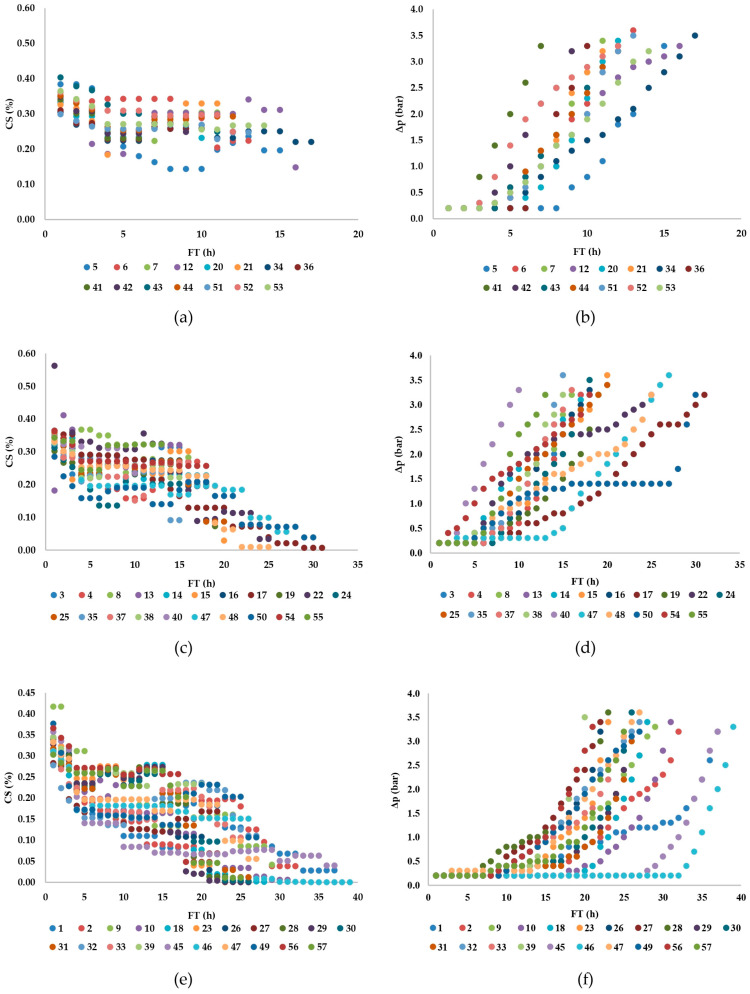
Suspension concentration of dosing filtration aid—CS (left)—and pressure differential—Δp (right)—during the filtration of sunflower oil with high (**a**,**b**), medium (**c**,**d**), and low (**e**,**f**) wax content.

**Figure 3 foods-13-02960-f003:**
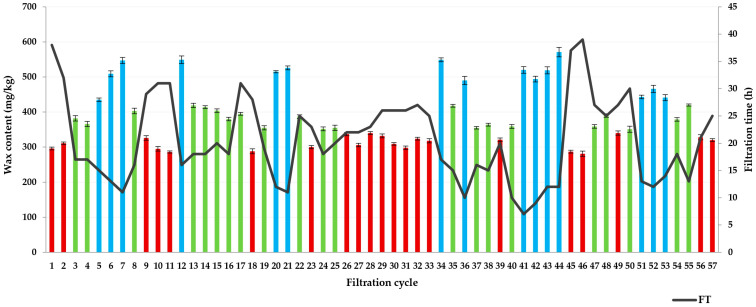
Comparative representation of wax content in sunflower oil before filtration and filtration time (■ low waxes; ■ medium waxes; ■ high waxes).

**Figure 4 foods-13-02960-f004:**
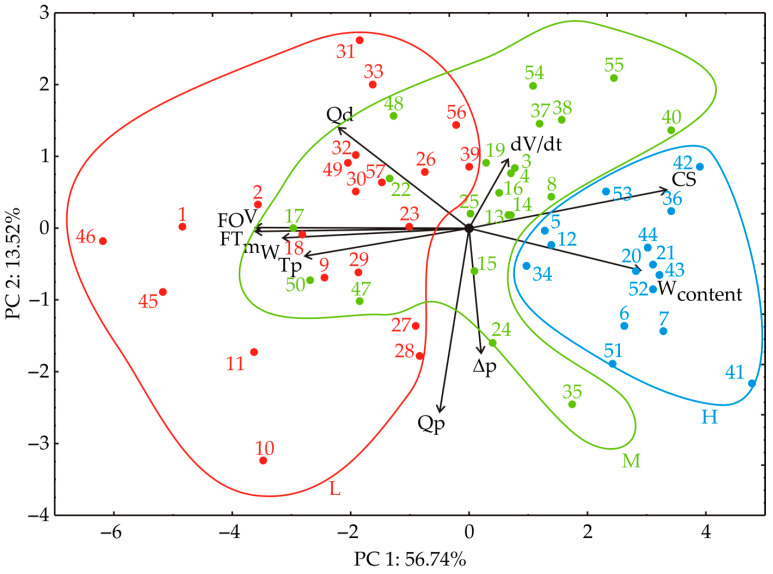
The PCA biplot diagram, depicting the relationships among W_content_, Qp, Qd, CS, and ∆p on FT, T_p const._, dV/dt, FO, V, and mW.

**Figure 5 foods-13-02960-f005:**
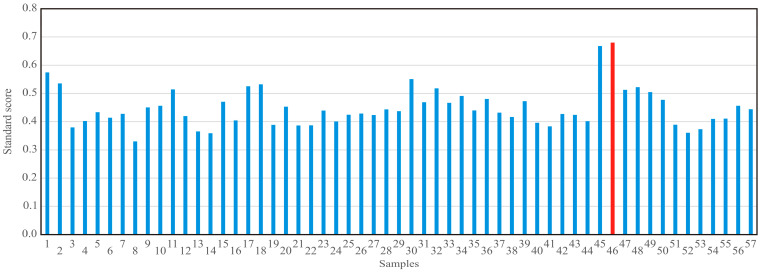
Standard scores for 57 samples. The red color in the column highlights the optimal sample, indicating that this sample has been identified as optimal sample.

**Figure 6 foods-13-02960-f006:**
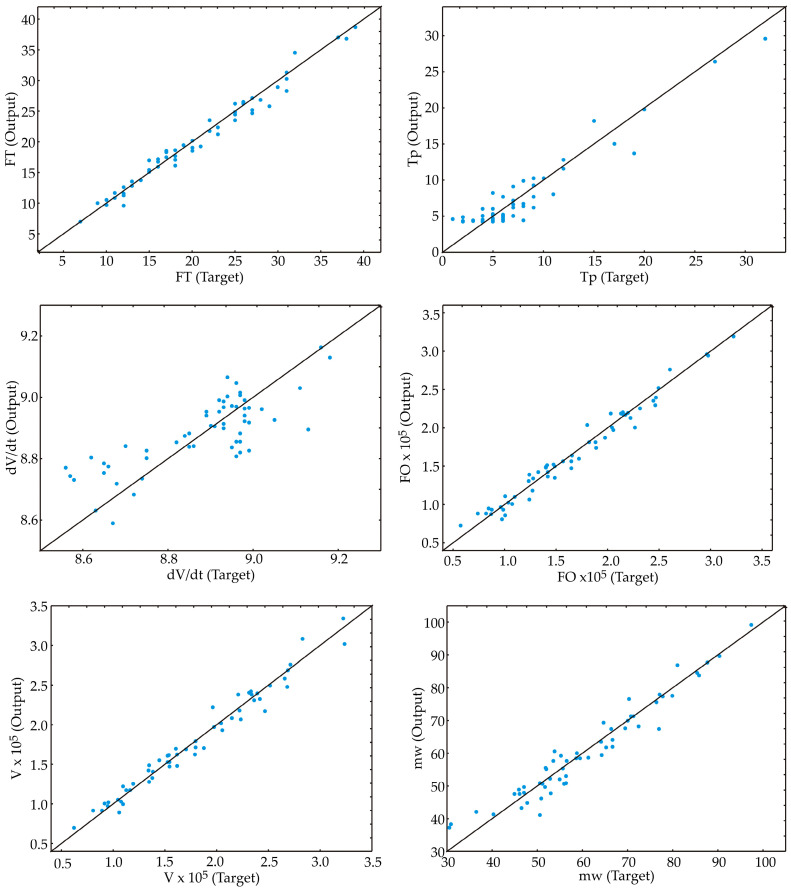
Experimental and predicted values obtained for FT (filtration time), T_p const_ (constant pressure period), dV/dt (flow rate of filtered oil), FO (quantity of filtered oil), V (volume of filtered oil), and mW (mass of separated waxes).

**Table 1 foods-13-02960-t001:** Process parameters monitored during the sunflower oil filtration process within winterization.

Filtration Cycle	Dosing Filtration Aid	Qp (kg)	Qd (kg)	CS (% *w/w*)	Δp (bar)	T_p const._ (h) ∆p = 0.2	dV/dt (m^3^/h)	FO (kg)	V (m^3^)	mW (kg)
1	FA-I	80	375	0.13	3.2	15	8.70	296,237	321.997	87.69
2	FA-I	80	375	0.14	3.2	12	8.85	260,581	283.240	81.04
3	FA-I	80	375	0.25	3.2	6	8.96	140,083	152.264	53.51
4	FA-I	80	350	0.23	3.2	7	9.05	141,554	153.863	51.81
5	FA-I	80	325	0.23	3.3	8	8.97	123,827	134.595	53.86
6	FA-I	80	325	0.30	3.6	3	8.68	103,829	112.858	52.85
7	FA-I	80	250	0.28	3.4	4	8.62	87,232	94.817	47.72
8	FA-I	80	375	0.31	3.2	6	8.63	126,976	138.017	51.17
9	FA-I	80	375	0.18	3.3	9	8.32	222,101	241.414	72.40
10	FA-I	125	325	0.13	3.4	17	8.66	247,125	268.614	69.44
11	FA-II	100	350	0.14	3.4	20	8.75	249,442	271.133	70.34
12	FA-I	80	375	0.27	3.3	4	8.67	127,570	138.663	70.04
13	FA-I	80	375	0.27	3.2	6	8.57	141,880	154.217	59.45
14	FA-I	80	375	0.26	3.2	4	8.56	141,797	154.127	58.70
15	FA-I	80	375	0.24	3.6	6	8.97	164,980	179.326	66.65
16	FA-I	80	375	0.25	3.3	7	8.97	148,604	161.526	56.47
17	FA-I	80	375	0.17	3.2	6	8.65	246,634	268.080	97.42
18	FA-I	80	375	0.17	3.4	19	8.99	231,483	251.612	66.67
19	FA-I	80	375	0.24	3.2	5	8.98	156,903	170.547	55.70
20	FA-I	80	250	0.26	3.3	5	8.95	98,810	107.402	50.89
21	FA-I	80	275	0.28	3.2	4	8.65	87,567	95.182	46.06
22	FA-I	80	450	0.24	3.2	4	8.58	197,439	214.608	76.41
23	FA-II	80	375	0.21	3.4	10	8.89	188,064	204.417	56.42
24	FA-I	100	325	0.21	3.5	7	8.93	147,808	160.661	52.03
25	FA-I	80	375	0.23	3.4	5	8.98	165,202	179.567	58.65
26	FA-II	80	375	0.21	3.2	8	8.98	181,800	197.609	61.27
27	FA-I	100	325	0.16	3.4	8	8.91	180,385	196.071	55.20
28	FA-I	100	350	0.20	3.6	6	8.93	189,063	205.503	64.28
29	FA-I	100	350	0.16	3.2	7	8.90	212,924	231.439	70.69
30	FA-I	60	400	0.19	3.6	12	8.98	214,737	233.410	66.35
31	FA-II	60	400	0.19	3.0	9	8.99	214,996	233.691	64.07
32	FA-I	60	400	0.19	3.4	7	8.85	219,891	239.012	71.24
33	FA-II	60	400	0.19	3.1	9	8.84	203,219	220.890	64.62
34	FA-II	80	375	0.25	3.5	5	8.97	140,305	152.505	77.03
35	FA-II	100	250	0.23	3.6	5	8.96	123,707	134.464	51.71
36	FA-I	60	225	0.26	3.3	6	8.94	82,212	89.361	40.28
37	FA-II	60	350	0.25	3.3	6	9.02	132,746	144.289	47.12
38	FA-II	60	325	0.25	3.2	5	8.94	123,411	134.142	44.92
39	FA-I	60	375	0.26	3.5	11	8.96	164,906	179.246	52.93
40	FA-I	60	275	0.29	3.3	2	9.16	84,266	91.593	30.25
41	FA-I	100	150	0.28	3.3	2	8.92	57,426	62.420	29.86
42	FA-I	60	250	0.28	3.2	2	8.93	73,913	80.340	36.51
43	FA-I	80	250	0.31	3.3	4	8.82	97,364	105.830	50.53
44	FA-I	80	300	0.29	3.2	2	8.72	96,254	104.624	54.96
45	FA-II	80	275	0.10	3.2	27	8.74	297,423	323.286	85.35
46	FA-I	80	375	0.12	3.3	32	8.99	322,503	350.547	90.41
47	FA-II	80	375	0.18	3.6	5	8.75	217,384	236.287	77.85
48	FA-II	60	375	0.20	3.2	5	8.93	205,337	223.192	79.88
49	FA-I	80	375	0.19	3.2	9	9.13	226,770	246.489	76.88
50	FA-I	100	350	0.15	3.2	4	8.86	244,400	265.652	85.78
51	FA-I	100	275	0.26	3.5	3	8.95	107,053	116.362	47.08
52	FA-I	100	275	0.31	3.3	2	9.11	100,530	109.272	46.48
53	FA-I	80	325	0.28	3.2	3	8.96	100,530	109.272	50.53
54	FA-II	60	375	0.28	3.2	1	8.97	148,548	161.465	56.00
55	FA-II	60	350	0.31	3.2	5	9.18	109,773	119.318	45.89
56	FA-I	60	375	0.24	3.3	8	8.92	172,409	187.401	56.38
57	FA-I	80	375	0.19	3.2	7	8.89	204,384	222.157	65.24

Qp—quantity of pre-coating filtration aid; Qd—quantity of dosing filtration aids; CS—concentration of dosing filtration aid; Δp—pressure differential; Τ_p const._—constant pressure period; dV/dt—flow rate of filtered oil; FO—quantity of filtered oil; V—volume of filtered oil; mW—mass of separated waxes.

**Table 2 foods-13-02960-t002:** Parameters calculated based on the measured data during the sunflower oil filtration process during winterization and results of oil analysis.

Filtration Cycle	W/A	Qp/A	Qd/A	Qaverage	FT	T_p const._/FT	∆p/∆t	mW/A	V/A
1	4.93	1.33	6.25	0.13	38	0.39	0.09	1.46	5366.61
2	5.18	1.33	6.25	0.14	32	0.38	0.14	1.35	4720.67
3	6.37	1.33	6.25	0.25	17	0.35	0.29	0.89	2537.74
4	6.10	1.33	5.83	0.23	17	0.41	0.33	0.86	2564.38
5	7.25	1.33	5.42	0.23	15	0.53	0.48	0.90	2243.24
6	8.48	1.33	5.42	0.30	13	0.23	0.38	0.88	1880.96
7	9.12	1.33	4.17	0.28	11	0.36	0.51	0.80	1580.29
8	6.72	1.33	6.25	0.27	16	0.38	0.32	0.85	2300.29
9	5.43	1.33	6.25	0.18	29	0.31	0.16	1.21	4023.57
10	4.92	2.08	5.42	0.13	31	0.55	0.22	1.16	4476.90
11	4.77	1.67	5.83	0.14	31	0.65	0.26	1.17	4518.88
12	9.15	1.33	6.25	0.27	16	0.25	0.28	1.17	2311.05
13	6.98	1.33	6.25	0.27	18	0.50	0.25	0.99	2570.29
14	6.90	1.33	6.25	0.26	18	0.94	0.21	0.98	2568.79
15	6.73	1.33	6.25	0.24	20	1.00	0.25	1.11	2988.77
16	6.33	1.33	6.25	0.25	18	0.22	0.3	0.94	2692.10
17	6.58	1.33	6.25	0.17	31	0.19	0.13	1.62	4468.01
18	4.80	1.33	6.25	0.17	28	0.68	0.38	1.11	4193.53
19	5.92	1.33	6.25	0.26	19	0.21	0.19	0.93	2842.45
20	8.58	1.33	4.17	0.28	12	0.50	0.52	0.85	1790.04
21	8.77	1.33	4.58	0.24	11	0.36	0.46	0.77	1586.36
22	6.45	1.33	8.33	0.28	25	0.24	0.11	1.27	3576.79
23	5.00	1.33	6.25	0.21	23	0.22	0.21	0.94	3406.96
24	5.87	1.67	5.42	0.21	18	0.39	0.28	0.87	2677.68
25	5.92	1.33	6.25	0.23	20	0.25	0.21	0.98	2992.79
26	5.62	1.33	6.25	0.21	22	0.36	0.19	1.02	3293.48
27	5.10	1.67	5.42	0.16	22	0.36	0.21	0.92	3267.84
28	5.67	1.67	5.83	0.2	23	0.26	0.15	1.07	3425.05
29	5.53	1.67	5.83	0.16	26	0.27	0.12	1.18	3857.32
30	5.15	1.00	6.67	0.19	26	0.46	0.21	1.11	3890.16
31	4.97	1.00	6.67	0.19	26	0.35	0.13	1.07	3894.86
32	5.40	1.00	6.67	0.19	27	0.26	0.17	1.19	3983.53
33	5.30	1.00	6.67	0.19	25	0.36	0.16	1.08	3681.50
34	9.15	1.33	6.25	0.25	17	0.29	0.26	1.28	2541.76
35	6.97	1.67	4.17	0.23	15	0.33	0.35	0.86	2241.07
36	8.17	1.00	3.75	0.26	10	0.60	0.65	0.67	1489.35
37	5.92	1.00	5.83	0.25	16	0.38	0.34	0.79	2404.82
38	6.07	1.00	5.42	0.25	15	0.27	0.29	0.75	2235.71
39	5.35	1.00	6.25	0.26	20	0.55	0.36	0.88	2987.43
40	5.98	1.00	4.58	0.29	10	0.20	0.41	0.50	1526.56
41	8.67	1.67	2.50	0.28	7	0.29	0.62	0.50	1040.33
42	8.23	1.00	4.17	0.28	9	0.22	0.50	0.61	1339.00
43	8.65	1.33	4.17	0.31	12	0.33	0.40	0.84	1763.84
44	9.52	1.33	5.00	0.29	12	0.17	0.38	0.92	1743.73
45	4.78	1.33	4.58	0.10	37	0.73	0.33	1.42	5388.10
46	4.68	1.33	6.25	0.12	39	0.82	0.47	1.51	5842.45
47	5.98	1.33	6.25	0.18	27	0.19	0.17	1.30	3938.12
48	6.50	1.00	6.25	0.20	25	0.20	0.12	1.33	3719.87
49	5.67	1.33	6.25	0.19	27	0.33	0.18	1.28	4108.15
50	5.85	1.67	5.83	0.15	30	0.13	0.06	1.43	4427.54
51	7.38	1.67	4.58	0.26	13	0.23	0.38	0.78	1939.37
52	7.77	1.67	4.58	0.31	12	0.17	0.32	0.77	1821.20
53	7.35	1.33	5.42	0.28	14	0.21	0.30	0.84	1821.20
54	6.32	1.00	6.25	0.28	18	0.06	0.16	0.93	2691.09
55	7.00	1.00	5.83	0.31	13	0.38	0.41	0.76	1988.64
56	5.47	1.00	6.25	0.24	21	0.38	0.24	0.94	3123.35
57	5.33	1.33	6.25	0.19	25	0.28	0.15	1.09	3702.61

W/A—wax content in oil before filtration per square meter of filtration area (mg/kg per 1 m^2^); Qp/A—amount of filtration aid applied per square meter of filtration area (kg/m^2^); Qd/A—quantity of dosing agent per square meter of filtration area (kg/m^2^); Qaverage—average concentration of dosing agent (%); FT—filtration time (h); T_p const._/FT—ratio of initial constant pressure time to filtration time; ∆p/∆t—pressure change per unit time (Pa/s); mW/A—wax mass per unit filtration area (kg/m^2^); V/A—volume of filtered oil per square meter of filtration area (m^3^/m^2^).

**Table 3 foods-13-02960-t003:** The optimized neural network models.

		Performance (r^2^)			Error			Train.	Error	Activation	
	Net. Name	Train.	Test	Valid.	Train.	Test	Valid.	algor.	func.	Hidden	Output
FT	MLP 7-5-1	0.977	0.976	0.968	0.653	0.595	1.044	BFGS 32	SOS	Exp.	Identity
T_p const._	MLP 7-5-1	0.927	0.905	0.721	1.252	1.215	3.886	BFGS 55	SOS	Logistic	Logistic
dV/dt	MLP 7-10-1	0.401	0.853	0.882	0.008	0.006	0.004	BFGS 52	SOS	Exp.	Tanh
FO	MLP 7-3-1	0.970	0.974	0.981	5.7 × 10^7^	4.0 × 10^7^	3.7 × 10^7^	BFGS 21	SOS	Tanh	Exp.
V	MLP 7-8-1	0.969	0.976	0.988	7.0 × 10^7^	4.3 × 10^7^	3.7 × 10^7^	BFGS 18	SOS	Exp.	Identity
mW	MLP 7-10-1	0.940	0.837	0.990	6.730	11.387	4.065	BFGS 32	SOS	Logistic	Exp.

FT—filtration time (h); Τ_p const._—constant pressure period; dV/dt—flow rate of filtered oil; FO—quantity of filtered oil; V—volume of filtered oil, mW—mass of separated waxes.

**Table 4 foods-13-02960-t004:** The “goodness of fit” tests for the developed neural network models.

	χ^2^	RMSE	MBE	MPE	SSE	AARD	r^2^	Skew	Kurt	Mean	StDev	Var
FT	1.424	1.183	0.205	4.613	77.372	51.474	0.967	0.364	0.162	0.205	1.175	1.382
T_p const._	3.291	1.798	0.009	33.730	184.293	81.881	0.924	0.236	0.181	0.009	1.814	3.291
dV/dt	0.015	0.120	−0.016	0.926	0.806	4.636	0.703	−1.796	8.260	−0.016	0.120	0.014
FO	1.1 × 10^8^	1.0 × 10^4^	924.573	5.806	5.9 × 10^9^	4.6 × 10^5^	0.966	0.027	−0.252	924.573	1.0 × 10^4^	1.1 × 10^8^
V	1.2 × 10^8^	1.1 × 10^4^	−19.177	5.164	7.0 × 10^9^	4.9 × 10^5^	0.959	0.203	0.206	−19.177	1.1 × 10^4^	1.2 × 10^8^
mW	14.269	3.744	0.206	5.450	796.652	165.644	0.953	0.075	0.061	0.206	3.772	14.226

FT—filtration time (h); Τ_p const._—constant pressure period; dV/dt—flow rate of filtered oil; FO—quantity of filtered oil; V—volume of filtered oil, mW—mass of separated waxes; χ^2^—reduced chi-square; RMSE—root mean square error, MBE—mean bias error; MPE—mean percentage error; SSE—sum of squared errors; AARD—average absolute relative deviation; r^2^—coefficient of determination.

## Data Availability

The original contributions presented in the study are included in the article/Appendix A, further inquiries can be directed to the corresponding author.

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
