# Peer review of "Implementation of Cellulose-Based Filtration Aids in Industrial Sunflower Oil Dewaxing (Winterization): Process Monitoring, Prediction, and Optimization"

_foods, 2024, doi:10.3390/foods13182960_

Round 1

Reviewer 1 Report

Comments and Suggestions for Authors

The study was about the usage of Cellulose-Based Filtration in Industrial Sunflower Oil Dewaxing, which is a very meaningful topic. The design of the research was scientific The article was well written. However, there are several parts that need to be revised:

Q1: In Abstract, please reduce the introduction of background and introduce more about your research including methods and results.

Q2: Check the grammar of Line 45-47.

Q3: State the main parameters of this method in 2.3.1

Q4: Explain the kind and source of oil ingredient used in this study.

Q5: Add the method of statistics analysis used in this study at the last part of Materials and Methods

Q6: How many times you repeat your study? Please state that in your paper.

Q7: Add the results of significant analysis in your results, thats essential.

Q8: Rewrite the conclusion, no need to introduce the background here again. Stress your results and the application prospect of your work.

Comments on the Quality of English Language

Recheck the grammar mistakes in your paper.

Author Response

The study was about the usage of Cellulose-Based Filtration in Industrial Sunflower Oil Dewaxing, which is a very meaningful topic. The design of the research was scientific. The article was well written. However, there are several parts that need to be revised:

AUTHORS: Thank you very much for a positive attitude toward our investigation.

Q1: In Abstract, please reduce the introduction of background and introduce more about your research including methods and results.

AUTHORS: Thank you for this comment. Authors have modified the abstract according to Reviewer's recommendations.

Q2: Check the grammar of Line 45-47.

AUTHORS: Thank you for this observation, the text was rearranged: “In addition to the traditional method, an alternative winterization process using solvents, most commonly n-hexane, is employed. The solvent reduces the viscosity of the oil, thereby facilitating the crystallization of waxes. The filtration process is conducted at lower temperatures (between –5ºC and –7ºC), which leads to a threefold costs reduction [7,8].”

Q3: State the main parameters of this method in 2.3.1

AUTHORS: Authors added the main parameters in the manuscript: “The method is based on the selective extraction of waxes, using their characteristics to crystallize and settle in oil at lower temperatures. Separated wax crystals are first purified with n-hexane at 1–3 °C for 8 h, then extracted using warm alcohol for 4–6 h.”

Q4: Explain the kind and source of oil ingredient used in this study.

AUTHORS: Authors added following information in the section 2.3. Oil Samples: “Industrial winterization (dewaxing) was carried out during 2019–2020, in Dijamant Ltd. company in Zrenjanin, Serbia. The obtained oil is produced from sunflower seeds (Helianthus annuus L.) grown in the territory of Vojvodina (north of the Republic of Serbia) in 2019.”

Q5: Add the method of statistics analysis used in this study at the last part of Materials and Methods.

AUTHORS: Thank you for this observation, section 2.7 was added to the text.

  1. 7. Statistical Analysis

Waxes content measurement was performed in triplicate (n = 3), while process parameters were recorded during industrial filtration and each trial is an individual cycle. Data from 57 industrial filtration cycles were analyzed using several statistical tests, including the Kruskal-Wallis test, the Kolmogorov-Smirnov two-sample test, the Wald-Wolfowitz runs test, and the Mann-Whitney U test, to identify significant differences between samples. Principal Component Analysis (PCA) was also conducted to structure and interpret the results. All statistical analyses were performed using Statistica 10.0 (StatSoft Inc., Tulsa, OK, USA).

Q6: How many times you repeat your study? Please state that in your paper.

AUTHORS: Thank you for this observation, total waxes content measurement was performed in triplicate (n = 3), while process parameters were recorded during industrial filtration and each trial is an individual cycle. The capacity of industrial filtration is large. The oil that comes to the filtration is of variable quality, it is not possible to completely reproduce the filtration cycle in the same way, so a higher number of cycles are monitored. Specific information is added in the section 2.7. Statistical Analysis.

Q7: Add the results of significant analysis in your results, that’s essential.

AUTHORS: Since industrial continuous production was examined in the article, it was not possible to monitor the process parameters in several trials, but individual trials were recorded as different filtration cycles. A total of 57 filtration cycles were monitored. Also, waxes content measurement was performed in triplicate (n = 3). Several statistical tests, including the Kruskal-Wallis test, the Kolmogorov-Smirnov two-sample test, the Wald-Wolfowitz runs test, and the Mann-Whitney U test, were used to identify significant differences between samples.

This information is added in the section 2.7. Statistical Analysis.

Q8: Rewrite the conclusion, no need to introduce the background here again. Stress your results and the application prospect of your work.

AUTHORS: Thank you for this comment. Authors have modified the conclusion according to reviewer's recommendations.

Reviewer 2 Report

Comments and Suggestions for Authors

My main concern with the present study is that the authors should have made a rough calculation of the process cost with and without cellulose -based filtration. Process efficiency may be higher using cellulose -based filtration but if the process cost remains high, this process will never be applied in industry.

My detailed comments follow the text sequence:

l.33-42: the Introduction begins specifically with winterization and then comes back to a more general concept, that of removal of undesirable compounds from crude oil i.e. degumming,

neutralization, bleaching, winterization and deodorization. The introduction should begin with this latter concept and concentrate on winterization.

l.37-39: this sentence I redundant

l.77: rewrite sentence in proper English.

l.78: on line 73 the authors mention the production of 4 kg of filter cake per ton of crude oil. On l. 78 they mention the production of 25% of filter cake. There is a huge difference between the two values. Obviously one of the two is incorrect !

l.98-99: sentence is redundant

l.102: was the study on winterization carried out at industrial scale ? If so mention plant name and location

l.104: show these two cellulose-based filtration aid steps on the flow diagram. I see only one such step

l.311: the application of PCA is not mentioned in the M+M section

Based on the above, I recommend major revision of the manuscript

Comments on the Quality of English Language

minor editing of English ir required.

Author Response

My main concern with the present study is that the authors should have made a rough calculation of the process cost with and without cellulose -based filtration. Process efficiency may be higher using cellulose -based filtration but if the process cost remains high, this process will never be applied in industry.

AUTHORS: Thank you for this observation, which is extremely important, and for the opportunity to explain your concern in detail. Detailed cost calculations are handled by experts from Dijamant Ltd. company, where the testing was conducted. The authors of this paper dealt with technological and scientific data significant for cost calculation. In the filtration process significant advantages of using these aids compared to perlite were recorded: for filtration of the same amount of oil, a lower quantity of cellulose-based filtration aids is required; it is possible to achieve a higher flow rate, which leads to a lower filtration time. Additionally, the time for cake drying is reduced. Furthermore, the oil content remaining in the cake obtained by filtration with cellulose aids is 20% lower compared to the cake with perlite. The resulting cake is easily removed from the leaves, while the filtration cake with perlite is difficult to remove, sticks to the leaves, and needs to be scraped off with shovels, which requires time, dirties the leaves, and poses a risk of damage. Since the leaves get dirty, uneven cake formation occurs, shortening the filtration cycle. When using perlite, the leaves need to be boiled 2-3 times a year to maintain their function, which is not the case with cellulose aids. Additionally, perlite is abrasive and causes frequent pump failures and pipeline blockages, which is not the case with cellulose aids. Moreover, the filtration cake with cellulose fibers can be further valorized as animal feed, while the cake with perlite is considered waste.

Considering these facts and savings in time, labor, equipment maintenance costs, reduced oil losses, by-product valorization, as well as the cost of the filtration aid itself and the optimal parameters obtained in this study, experts from Dijamant Ltd. concluded that replacing perlite with cellulose-based filtration aids in their production plant is economically justified.

My detailed comments follow the text sequence:

AUTHORS: Thank you very much for a positive attitude toward our investigation.

l.33-42: the Introduction begins specifically with winterization and then comes back to a more general concept, that of removal of undesirable compounds from crude oil i.e. degumming,

neutralization, bleaching, winterization and deodorization. The introduction should begin with this latter concept and concentrate on winterization.

AUTHORS: Thank you for this comment. Authors have modified the Introduction section according to Reviewer's recommendations.

l.37-39: this sentence I redundant

AUTHORS: The sentence is removed from the article.

l.77: rewrite sentence in proper English.

AUTHORS: Thank you for this observation, the sentence was rearranged, according to the Reviewer’s comment: “According to the Statistical Office of the Republic of Serbia data, the annual production of refined sunflower oil in Serbia averages between 160000 and 180000 tons, generating approximately 640 to 720 tons of filter cake, practically that amount of “waste”.

l.78: on line 73 the authors mention the production of 4 kg of filter cake per ton of crude oil. On l. 78 they mention the production of 25% of filter cake. There is a huge difference between the two values. Obviously one of the two is incorrect!

AUTHORS: Thank you for this observation, the authors made a mistake in the calculation. Namely, 4 kg of filter cake is generated per ton of refined oil, which indicates that between 640 and 720 tons of filter cake are generated for the annual production of refined sunflower oil of 160000-180000 t. Authors have modified this information in the manuscript.

l.98-99: sentence is redundant.

AUTHORS: The sentence is removed from the article.

l.102: was the study on winterization carried out at industrial scale? If so mention plant name and location.

AUTHORS: Yes, the study was carried out at industrial scale in Dijamant Ltd. company in Zrenjanin, Serbia. Authors added this in the manuscript in the section 2.3. Oil Samples.

l.104: show these two cellulose-based filtration aid steps on the flow diagram. I see only one such step.

AUTHORS: The suspension of filtration aids used for pre-coat (FA-P) and dosing (FA-I or FA-II) is prepared in one tank, only different filtration aids are dosed depending on the filtration phase. So, if the filtration aid FA-P is used during filtration, it is dosed at place marked with “cellulose-based filtration aids” on the diagram, also if FA-I or FA-II is used, it is dosed at the same place. Also, one tank supplies both horizontal pressure leaf filters as they work alternately to ensure a continuous process.

l.311: the application of PCA is not mentioned in the M+M section

AUTHORS: Thank you for this observation, section 2.7 was added to the text.

  1. 7. Statistical Analysis

Waxes content measurement was performed in triplicate (n = 3), while process parameters were recorded during industrial filtration and each trial is an individual cycle. Data from 57 industrial filtration cycles were analyzed using several statistical tests, including the Kruskal-Wallis test, the Kolmogorov-Smirnov two-sample test, the Wald-Wolfowitz runs test, and the Mann-Whitney U test, to identify significant differences between samples. Principal Component Analysis (PCA) was also conducted to structure and interpret the results. All statistical analyses were performed using Statistica 10.0 (StatSoft Inc., Tulsa, OK, USA).

Based on the above, I recommend major revision of the manuscript

AUTHORS: Thank you for all your comments, we revised the document according to the Reviewers’ observation and hope that you’ll find the revised satisfactory to be accepted for publication in the Foods journal.

Round 2

Reviewer 2 Report

Comments and Suggestions for Authors

My main concern was not addressed. The authors in collaboration with the company, should sit down and make a rough calculation of the proposed process cost. If the cost is high, the proposed process will never be applied to industry.

Comments on the Quality of English Language

Minor editing of English required

Author Response

The authors regret that the previous explanation was not sufficient to alleviate your concerns, so we will make an effort to further clarify the existing worries.

To begin with, a brief overview of the prices of the filtration aids used in the 57 filtration cycles examined is provided:

  • Conventional non-cellulosic filtration aid

Depending on the wax content of the oil being winterized, the price ranges from a minimum of 96.00 RSD/t of filtered oil (FO) to a maximum of 168.00 RSD/t FO, with an average price of 130.00 RSD/t FO for all 57 filtrations (1.00 euro = 118.00 RSD). 

  • Cellulose-based filtration aid FA-I

Depending on the wax content of the oil being winterized, the price ranges from a minimum of 40.19 RSD/t of filtered oil (FO) to a maximum of 282.10 RSD/t FO, with an average price of 89.29 RSD/t FO for all 57 filtrations. 

  • Cellulose-based filtration aid FA-II

Depending on the wax content of the oil being winterized, the price ranges from a minimum of 57.84 RSD/t of filtered oil (FO) to a maximum of 375.00 RSD/t FO, with an average price of 129.29 RSD/t FO for all 57 filtrations.

Annually, the production plant of the company Dijamant Ltd. in Zrenjanin produces about 57,000.00 t of winterized oil, using cellulose-based filtration aids. Approximately 300 t of filter cake are generated. The disposal costs for exhausted cellulose-based filtration aids are 0.00 euros, while the cake is sold, and the profit from this is not included in the calculation. If conventional non-cellulosic aid are used, the disposal cost amounts to 22,500.00 euros.

The maintenance costs of equipment when working with cellulose-based filtration aid on an annual basis amount to 67,600.00 euros, while when dealing with conventional non-cellulosic filtration aid, it amounts to 132,000.00 euros.

As mentioned in the previous response, when discussing cost-efficient and economical filtration, it is important to note that compared to conventional non-cellulosic (mineral) filtration aids, the purchasing costs of cellulose are usually higher. This fact, however, is more than compensated for by the positive properties of cellulose based filtration aids. Its low wet cake density results in considerably less filtration aid being consumed than with mineral filtration aids. In addition, product losses are reduced due to high permeability, and disposal of filter cake does not incur any costs. Moreover, using cellulose-based aids substantially reduces plant maintenance costs. To conclude, filtration with cellulose-based filtration aids is much more economical than with conventional filter aids. Due to all the aforementioned facts and the presented calculations, the company Dijamant Ltd. in Zrenjanin has decided to use this filtration aids, which is in operation in their production plant. The authors also noted in the conclusion of the manuscript that economic justification has been achieved (Lines 442-444).

It is also important to emphasize that this data is only valid for this single production plant, the manner of equipment usage, the specific quality of oil being refined, working conditions, etc. If any of the aforementioned changes occur in this production plant, or if industrial winterization is performed in a different plant with different equipment, crude oil, etc., this data will not remain the same.